# INTERPRETING AGE PREDICTIONS FROM BRAIN MAPS VIA DEEP NEURAL ACTIVATIONS AND TENSOR DECOMPOSITION

## ABSTRACT

Deep learning models, while effective, often lack transparent interpretability, especially in critical areas like healthcare. This paper introduces a novel approach to interpret 3D convolutional neural networks trained to estimate one or more of the individual's clinically relevant attributes from 3D brain maps. In contrast to interpretability methods commonly applied to object classification in images, such as gradient approaches like GradCAM, which must rely on per-instance explanations due to the spatial variation of object location, brain maps have a common spatial registration and we propose to compute explanations at the dataset-level. After organizing the internal activations of 3D convolutional neural network across the training dataset into a tensor, we use a constrained tensor decomposition to reveal the key spatial patterns that highlight the specific regions of the brain the model focuses on during its predictions. We use reconstruction error to guide the selection of the rank of the tensor decomposition, and fit linear models to relate the decomposition of activations to the original target attributes. We apply the method to network trained to estimate an individual's chronological age using brain maps of volume and stiffness computed from magnetic resonance imaging (MRI) and T1-weighted magnetic resonance elastography (MRE) scans, respectively. The tensor decomposition's spatial factors predominantly emphasize areas of the brain known to vary with aging. Additionally, the linear model fit to the decomposition has only a slight decrease in performance. The proposed decomposition technique provides a mechanism to interpret convolutional models applied to brain maps, and offers potential insights into the age-related structural changes in the brain.

## 1 INTRODUCTION

Deep neural networks (DNNs) have been extensively applied in medical imaging domains for both diagnosis and prognosis—achieving human-level performance in the early diagnosis of numerous diseases (Singh et al., 2020; Zeineldin et al., 2022; Dyrba et al., 2021; Taşçı, 2023), especially in specializations such as neuroradiology, offering impressive performance in tasks such as disease classification and tumor analysis (). For neuroimaging, magnetic resonance imaging (MRI) is an indispensable tools for delving into the intricate terrains of brain structure and function (Chen et al., 2022; Zhang et al., 2021) and assisting in medical image diagnoses and prognoses (Qian et al., 2023).

The ability of DNNs to solve computer vision tasks has been attributed to the similarity of DNN processing with the hierarchical processing in the human visual system (Farahani et al., 2022). Nonetheless, even if DNNs process images in a manner analogous to the visual system, the overarching challenge remains the interpretability of these models (Qian et al., 2023). Models that are not interpretable are not going to used. Additionally, neuroscientists and neurologists stand to gain trust knowledge by better understanding the patterns detected by a model.

In this paper, we introduce a novel approach to interpret the predictions of 3D convolutional neural networks trained on brain maps derived from magnetic resonance imaging (MRI) and magnetic resonance elastography (MRE) scans to estimate one or more of the individual's clinically relevant attributes. In contrast to interpretability methods commonly applied to object classification in images, such as gradient approaches like GradCAM (Selvaraju et al., 2017), which must rely on per-instance

explanations due to the spatial variation of object location, brain maps have a common spatial registration. Likewise, the internal activations of 3D convolutional neural network organized into a 4D tensor per brain map, and a 5D tensor across the dataset. The main contributions are:

- The design of a constrained and regularized tensor decomposition that when applied to the tensor of internal, non-negative activations of a convolutional neural network reveals the key spatial intensity patterns highlighting the specific regions of the brain the model focuses on during its predictions.
- A rank selection approach informed by scree plot analysis of the reconstruction error.
- An approach for fitting a linear model to relate the decomposed activations and the original target variable.
- A detailed empirical analysis of the method when applied to neural networks that estimate an individual's chronological age from brain volume and brain stiffness measures. The spatial patterns underlying the activations predominantly emphasize areas of the brain known to vary with aging.

This methodology not only leverages the strengths of deep learning in feature extraction but also imbues the model with enhanced interpretability, thereby contributing a novel paradigm to the field of explainable artificial intelligence in medical imaging. Additionally, the results offer potential insights into the neurobiological processes underlying age-related changes in the brain.

## 2  RELATED WORK

This section reviews key contributions addressing the interpretability of deep learning models, specifically within the context of 3D imaging data and neuroimaging.

Countering the opacity of deep learning models, a plethora of interpretability methodologies have surfaced. Among these, methods like Class Activation Maps (CAM) (Zhou et al., 2016) and Gradient-weighted Class Activation Mapping (Grad-CAM) (Selvaraju et al., 2017) stand out. Grad-CAM offers 'visual explanations' for CNN-based model decisions, enhancing their transparency. The technique taps into gradients flowing to the final convolutional layer, producing localization maps that emphasize critical image regions for decision-making. Other techniques that also exploit activation maps such as Respond-weighted Class Activation Mapping (Respond-CAM) (Zhao et al., 2018) and Regression Activations Maps (RAM) (Mishra et al., 2020), visualize the regions in 3D biomedical imaging data that are relevant for making specific predictions. With a focus on post-hoc relevance techniques in neuroimaging data, Farahani et al. (2022) introduce a methodology for evaluating the reliability of explainability techniques, especially in the context of DNNs.

In a different avenue of interpretability for deep neural networks, Deep Feature Factorization (DFF) is a method that aims to localize similar semantic concepts within an image or a set of images by exploiting the learned features of deep convolutional neural networks (Collins et al., 2018). This technique leverages nonnegative matrix factorization to generate hierarchical cluster structures in feature space, highlighting semantically similar regions across multiple images. Similarly, tensor decomposition, and especially the Nonnegative Tucker Decomposition (NTD), serves as a potent instrument for extracting nonnegative, parts-based, and inherently meaningful latent components from high-dimensional tensor data (Zhou et al., 2015). However, the application of NTD to the activations of a trained model, as proposed in this work, has not been reported previously.

## 3  METHODOLOGY

### 3.1  ACTIVATION MAPS

We consider the internal representations of a 3D convolutional network. Given an input 3D brain map $\mathcal{X} \in \mathbb{R}^{h \times w \times d}$, the model generates a 4D tensor of activations $\mathcal{A}^{(l)} \in \mathbb{R}^{c_l \times h_l \times w_l \times d_l}$ at each layer $l$, where $h_l < h, w_l < w, d_l < d, l \in \{1, 2, \ldots, L\}$, consisting of $c_l$ channels of 3D feature activation maps at the $l$-th layer. In this high-dimensional feature space, each channel represents a specific feature that the network has learned through training. Each 3D feature map has a similar spatial arrangement as the input, but with lower resolution. The last convolutional layer $L$ of a model

represents the most complex features of the input volume, but at the coarsest spatial resolution. This feature space is especially crucial in medical contexts, where the learned features correspond to patterns in the input ranging from the edge contours of anatomical structures to texture variations often indicative of pathological states (Litjens et al., 2017).

## 3.2 TUCKER DECOMPOSITON

Tucker decomposition is a form of higher-order tensor factorization that generalizes matrix factorizations, such as the singular value decomposition (SVD), to tensors (Kolda & Bader, 2009). The primary objective is to achieve a compressed representation that encapsulates the crucial structural elements and multi-way interactions in the tensor (Kroonenberg & De Leeuw, 1980). The decomposition approximates the original tensor as a core tensor that is smaller in size, along with a set of factor matrices corresponding to each mode or dimension of the tensor. In general, by Tucker decomposition, a tensor $\mathcal{X} \in \mathbb{R}^{I_1 \times I_2 \times \cdots \times I_N}$ is approximated as $\mathcal{X} \approx \hat{\mathcal{X}} = \mathcal{G} \times_1 \mathbf{U}^{(1)} \cdots \times_N \mathbf{U}^{(N)}$, where $\mathcal{G} \in \mathbb{R}^{R_1 \times R_2 \times \cdots \times R_N}$ is the low rank core and $\mathbf{U}^{(k)} \in \mathbb{R}^{R_k \times I_k}$ are the projections factors with $k \in \{1, \ldots, N\}$. As an optimization problem, Tucker decomposition is formulated as

$$\min_{\mathcal{G}, \mathbf{U}^{(1)}, \ldots, \mathbf{U}^{(N)}} \|\mathcal{X} - \mathcal{G} \times_1 \mathbf{U}^{(1)} \cdots \times_N \mathbf{U}^{(N)}\| \tag{1}$$

Various algorithms exist for performing Tucker decomposition, one of which is the Higher-Order Orthogonal Iteration (HOOI) algorithm (Kofidis & Regalia, 2002). HOOI is essentially an extension of traditional SVD techniques but applied to tensors. It iteratively optimizes orthogonal factor matrices for each mode by minimizing the reconstruction error between the original and approximated tensors (De Lathauwer et al., 2000). Other approaches have also included optimization-based algorithms that can incorporate constraints like non-negativity, thus enhancing the interpretability of the decomposed components (Zhou et al., 2015).

## 3.3 TUCKER DECOMPOSITION ON ACTIVATION MAPS

The activation map for the last convolutional layer for the $i$th subject is denoted $\mathcal{A}_i^{(L)} \in \mathbb{R}^{c_L \times h_L \times w_L \times d_L}$, $i \in \{1, \ldots, n\}$. We can reshape this tensor to a matrix by vectorizing the spatial features, i.e, $\mathbf{A}_i^{(L)} \in \mathbb{R}^{c_L \times v_L}$, where $v_L = (h_L \cdot w_L \cdot d_L)$ are the number of voxels at the $L$th layer. Concatenating the activation matrices for a set of $n$ subjects yields the tensor $\mathcal{A} \in \mathbb{R}^{n \times c_L \times v_L}$. This 3-way tensor composed of the activations maps of features across subjects captures all of the information extracted by the deep neural network.

To decompose this high-dimensional tensor, we utilize the aforementioned Tucker decomposition. Specifically, we solve

$$\min_{\mathcal{G}, \mathbf{U}^{(1)}, \mathbf{U}^{(2)}, \mathbf{U}^{(3)}} \|\mathcal{A} - \mathcal{G} \times_1 \mathbf{U}^{(1)} \times_2 \mathbf{U}^{(2)} \times_3 \mathbf{U}^{(3)}\| \tag{2}$$

$$\text{s.t.} \quad \mathbf{U}^{(3)} \geq 0,$$
$$\mathcal{G}_{ijk} \geq 0 \text{ for } i = k,$$
$$\mathcal{G}_{ijk} = 0 \text{ for } i \neq k,$$

where $\mathcal{G} \in \mathbb{R}^{R_1 \times R_2 \times R_1}$, $\mathbf{U}^{(1)} \in \mathbb{R}^{R_1 \times n}$, $\mathbf{U}^{(2)} \in \mathbb{R}^{R_2 \times c_L}$, $\mathbf{U}^{(3)} \in \mathbb{R}^{R_1 \times v_L}$, $i, k \in \{1, \ldots, R_1\}$, and $j \in \{1, \ldots, R_2\}$. Note that the core $\mathcal{G}$ is composed of stacked square matrices across the second dimension.

### 3.3.1 SOLVING THE TUCKER DECOMPOSITION

The optimization problem in (2) can be effectively solved using a hybrid algorithm that combines Hierarchical Alternating Least Squares (HALS) (Cichocki et al., 2007) for the projection factors and Fast Iterative Shrinkage Thresholding Algorithm (FISTA) (Beck & Teboulle, 2009) for the core tensor. More specifically, each mode of the tensor is associated with a projection matrix. To compute these matrices, the HALS algorithm provides a computationally efficient and stable way to update each factor matrix while keeping others fixed. The algorithm iteratively refines the factors in a way that minimizes the reconstruction error, measured by the Frobenius norm. To compute the core, we use FISTA, which is designed to solve convex optimization problems which can be adapted to deal

with sparsity constraints of arbitrary shape. The implementation is based on the non-negative Tucker decomposition developed by Tensorly (Kossaifi et al., 2016).

## 3.4 INTERPRETATION OF TUCKER DECOMPOSITION

The result obtained after solving (2) consists of the core $\mathcal{G}$ and the projection factors $\mathbf{U}^{(1)}$, $\mathbf{U}^{(2)}$, and $\mathbf{U}^{(3)}$. In particular, we are interested in the first factor $\mathbf{U}^{(1)}$, which can be interpreted as the subject-specific loadings, and the third factor $\mathbf{U}^{(3)}$, which can be interpreted as spatial pattern across the voxels. It's worth noting that by imposing constraints on (2), we aim to achieve two goals:

- obtain interpretable spatial factors $\mathbf{U}^{(3)}$, which highlight the spatial extent of relevant voxels, by imposing non-negativity constraints and sparsity-inducing $\ell_1$-norm regularization; and

- ensure that the projection factor $\mathbf{U}^{(1)}$ and $\mathbf{U}^{(3)}$ are related by a one-to-one mapping, meaning that a set of subject-specific loadings are associated uniquely to the set of relevant voxels, by imposing diagonal faces of the core tensor.

Each column of the spatial factor $\mathbf{U}^{(3)}$ can be reshaped back into a spatial map of dimension $h_L \times w_L \times d_L$ to highlight a regions of interest (RoI) that corresponds to the subject-specific loadings in a column of $\mathbf{U}^{(1)}$. To visualize these RoIs in the resolution of the original brain maps, we upsample the RoIs using a bilinear transformation.

## 3.5 RANK SELECTION

Rank selection is a pivotal step in a Tucker decomposition, influencing both the quality and interpretability of the tensor factorization (Kolda & Bader, 2009). Several methods are available for determining the optimal ranks for each mode of the tensor. For this work we focus on two criteria:

- **Scree Plots** of the singular values which offer a graphical approach for rank selection. A sudden drop in a singular value magnitude can serve as an indicator for an appropriate cut-off rank (Jolliffe, 2002).

- **Elbow method** (Thorndike, 1953) of the normalized reconstruction error across various combinations of ranks. The combination of ranks that yield factors with minimal reconstruction error on held-out data can serve as an indicator for a suitable rank.

By leveraging these two methods, we determine the appropriate ranks for Tucker decomposition, thereby ensuring a more interpretable and efficient reduced-order model.

### 3.5.1 RANK SELECTION APPROACH

We analyze scree plots and elbow of the reconstruction error to determine an appropriate rank:

- For scree plots, we compute the singular value decomposition for the modes 1, 2, and 3 matricization of the activations collected for the training set $\mathcal{A}_{\text{train}}$.

- For the reconstruction error, we solve problem (2) considering activations for training set $\mathcal{A}_{\text{train}}$ and validation set $\mathcal{A}_{\text{val}}$, where the ranks $(R_1, R_2)$ are all possible combinations of $R_1 \in \{2, \ldots, 20\}$ and $R_2 \in \{2, \ldots, 20\}$. Algorithm 1 in the Appendix summarizes our approach.

## 3.6 COMPUTATION OF THE RoIs

After finding an adequate rank pair $(R_1, R_2)$, we solve problem 2 adding and sparsity constraint on the projection factor $\mathbf{U}^{(3)}$ such that the RoIs highlight regions of the brain that emphasize the physical measurement to correlate linearly with our target. To achieve this, we evaluate multiple regularization parameters $\lambda$, and use the approximated tensor $\hat{\mathcal{A}}_{\text{train}} = \mathcal{G}_{\text{train}} \times \mathbf{U}^{(1)}_{\text{train}} \times \mathbf{U}^{(2)}_{\text{train}} \times \mathbf{U}^{(3)}_{\text{train}}$ to train a linear model to predict the target $\mathbf{y}_{\text{train}}$ of the approximation after performing global averaging across the voxels $\hat{\mathbf{A}}_{\text{train}} = \text{global\_average}(\hat{\mathcal{A}}_{\text{train}}) = \frac{1}{v_L} \sum_{k=1}^{v_L} [\hat{a}_{\text{train}}(i, j, k)]_{i=1, j=1}^{n_{\text{train}}, c_L} \in \mathbb{R}^{n_{\text{train}} \times c_L}$. We apply the same transformation to the validation set $\hat{\mathbf{A}}_{\text{val}} = \text{global\_average}(\hat{\mathcal{A}}_{\text{val}}) \in \mathbb{R}^{n_{\text{val}} \times c_L}$ such that we

can evaluate the trained linear model. We then pick the regularization parameter $\lambda_{best}$ that gets the smallest MAE in predicting the attribute $\mathbf{y}_{val}$ in the validation set. Algorithm 2 in the Appendix synthesizes our approach.

## 4 EXPERIMENTS

### 4.1 IMPLEMENTATION DETAILS

#### 4.1.1 DATA

The total data set consists of 279 subjects with ages ranging from 5 to 81 years (116 males, 163 females). The data were compiled from 8 studies with similar MRI protocols, which included an magnetic resonance elastography (MRE) scan to generate maps of the mechanical properties (stiffness and damping ratio) of brain tissue and a high resolution T1-weighted anatomical scan to generate volume brain maps of the gray matter. From these scans, for each individual in our data set, we use either the brain stiffness or the volume brain maps registered by study-specific templates to the MNI152 atlas. From the data set of 279 subjects, roughly 20% (57 subjects) were reserved for the test group, and the remainder (222 subjects) were assigned to the training (166 subjects) and validation (56 subjects, 20% of the total data set) groups. This data set division was performed at random preserving the distribution of ages in each split. Since all the subjects considered in this study were cognitively normal, their brain age was assumed to be equal to their chronological age. This data was collected under regulatory approval and details are reported in existing publications—references removed for anonymity.

#### 4.1.2 MODEL

The architecture of the model used in this work is rooted in the 3D ResNet architecture proposed by He et al. (2016a). The 3D ResNet architecture extends the capabilities of 2D ResNets by adding a volumetric dimension to the feature space, making them particularly suited for medical imaging tasks like MRI or MRE scans. Specifically, we adapt the residual units to 3D to enable the handling of volumetric data commonly found in medical imaging, such as Magnetic Resonance Imaging (MRI) and Computed Tomography (CT) scans. A distinguishing feature of our model is the use of pre-activation residual units (He et al., 2016b), which introduce activation functions (e.g., ReLU) before the convolutional layers within each residual block. The pre-activation design, consisting of a Batch Normalization (BN) layer followed by a ReLU activation function and a 3D Convolution (Conv3D) layer, helps in reducing overfitting and accelerating convergence during training(He et al., 2016b). Our 3D ResNet consists of 5 blocks with 3, 4, 6, 4, and 3 residual units and 8, 16, 32, 64 and 128 channels, respectively in each block.

#### 4.1.3 TRAINING

In this study, two separate 3D ResNet models were trained on stiffness and volume brain maps, respectively, capitalizing on the architecture's demonstrated efficacy in high-dimensional data analysis tasks (He et al., 2016a). The dataset was divided into a 60/20/20 partition for training, validation, and test sets, as described in Section 4.1.1. Hyperparameters such as learning rate, batch size, weight decay, and momentum were optimized using the Tree-structured Parzen Estimator algorithm (Bergstra et al., 2013). Stochastic gradient descent was employed as the optimization algorithm, and the models were implemented using the PyTorch framework (Paszke et al., 2019). Quantitative assessment was conducted using Mean Absolute Error (MAE) and Mean Squared Error (MSE) as metrics. The model trained on stiffness maps yielded an MAE of 3.89 and an MSE of 27.08, while the volume-trained model achieved an MAE of 3.53 and an MSE of 23.83. These initial results suggest promising avenues for further research and comparative evaluation.

### 4.2 AGE PREDICTIONS FROM MRE DATA

Magnetic resonance elastography (MRE) is an innovative imaging technique that goes beyond conventional MRI to offer quantitative measurements of tissue stiffness and other biomechanical properties Muthupillai et al. (1995); Mariappan et al. (2010). Given the brain's viscoelastic nature, MRE has been explored as a potential biomarker for various neurological conditions and natural aging

processes (Murphy et al., 2013). Recent research has indicated that MRE-derived measurements, such as brain tissue stiffness, show correlations with age, providing an avenue for age prediction (Schwarb et al., 2016; McIlvain et al., 2022). These measures are increasingly being seen as complementary to volumetric measures obtained from conventional MRI for predicting age (Johnson et al., 2016).

### 4.2.1 RANK SELECTION

The scree plots for mode-1 and mode-3 unfoldings in Figure 1(a) and 1(c), respectively, show that after rank 7 the drop in magnitude of the singular values is not significant. Similarly, the reconstruction error in Figure 1(d) also demonstrates that at rank 7 the error stops decreasing substantially. The scree plot for mode-2 shows an abrupt reduction of two orders of magnitude at rank 4, and three orders of magnitude at rank 20. Based on the behavior of the scree plots and reconstruction error, we chose $R_1 = 7$ and $R_2 = 20$.

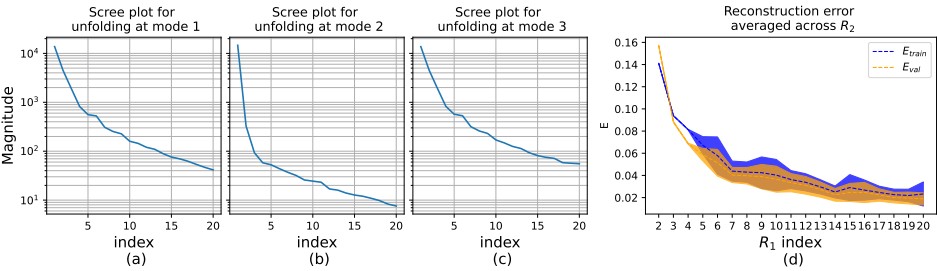

Figure 1: Scree plots and reconstruction error for MRE data. Scree plots for mode-1 $(a)$ and mode-3 $(c)$ unfoldings show a reduction in the magnitude of the singular values of almost two orders of magnitude at rank 10. The reduction in magnitude for mode-2 $(b)$ unfolding is even more drastic, decreasing more than two orders of magnitude at rank 4. The reconstruction error plots $(d)$ for the validation and training sets show a considerable drop at rank $R_1 = 7$.

### 4.2.2 SUBJECT-SPECIFIC LOADINGS $\mathbf{U}^{(1)}$, REGIONS OF INTEREST AND AGING

The relationship between brain stiffness and aging has been an emerging area of interest, particularly investigated using MRE techniques. Sack et al. (2009) presented one of the earlier works demonstrating that brain stiffness decreases with age, particularly in the frontal lobes. Similarly, Coelho & Sousa (2022) reported brain stiffness changes for two populations: the first group comprised of subjects between 20 and 60 years showed the most significant changes in sensorimotor regions, and the second group which included subjects between 60 and 90 exhibited the most prominent changes in temporal and occipital lobes. Overall, these studies provide critical insights into the relationship between brain stiffness and aging, suggesting potential utility in clinical diagnosis and treatment monitoring.

The set of seven plots in Figure 2 explores the relationship between age and loadings $\mathbf{U}^{(1)}$ related to brain stiffness. Interestingly, for children under the age of 10, the loadings are nearly zero across all plots, indicating minimal changes in brain stiffness during early childhood. However, the situation changes markedly during adolescence and adulthood. While RoIs 3, 4 and 5, (Figure 2(c), Figure 2(d), Figure 2(e)) illustrate a linear relationship between age and loadings, the remaining four plots depict a complex behavior, highlighting that RoIs 1, 2, 6 and 7 show a non-linear dependence between brain stiffness and age.

Figure 3 shows a combined view of the regions of interest. Each voxel in the 3D view is colored depending on the strongest magnitude across all RoIs. Interestingly, RoI 5, which covers frontal and parietal lobes mostly, and RoI 6, which is associated to the occipital lobe, account for most of the spatial locations in the brain. These regions corroborate the findings reported by Coelho & Sousa (2022) and Sack et al. (2009).

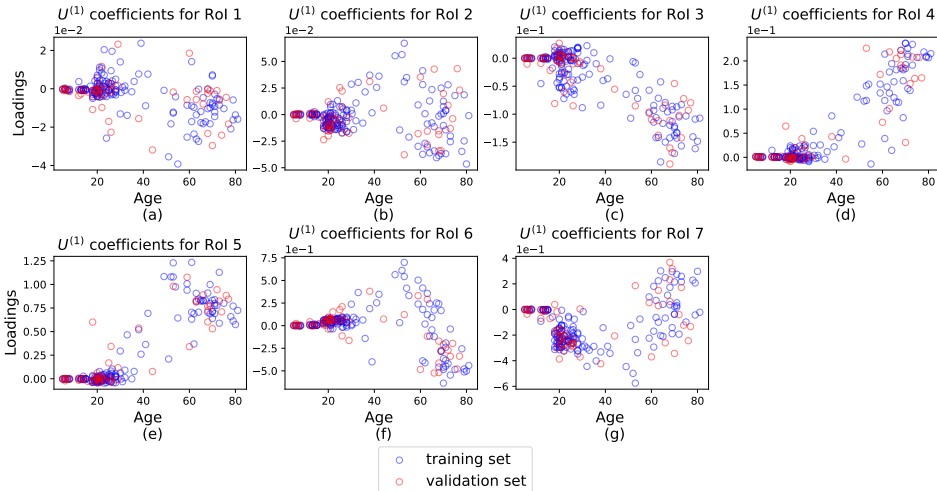

Figure 2: Loadings for RoIs. Relationships between age and loadings shows linear behavior for RoIs 3, 4 and 5, and non-linear behavior for RoI 1, 2, 6, and 7. Minimal variations in stiffness are shown for younger population, while older population exhibits significant changes.

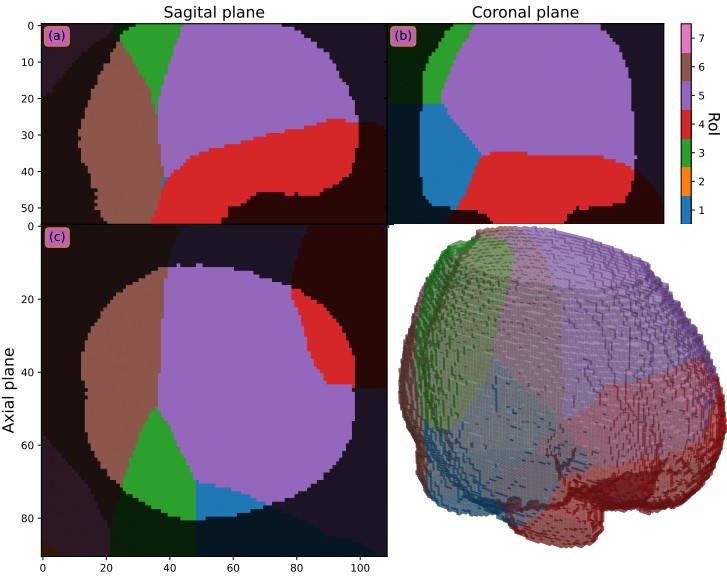

Figure 3: Combined Regions of Interest for Stiffness. Colored regions highlight which RoI has the strongest influence in a specific spatial location of the brain. RoIs 5 and 6 which are the most extended focus mostly on the parietal, frontal and occipital lobes.

## 4.3 AGE PREDICTIONS FROM MRI DATA

Age prediction based on brain morphometry has become an important area of research in both clinical neurosciences and radiology (Cole et al., 2015). Magnetic resonance imaging (MRI) is commonly used to capture high-resolution images of the brain, which can then be segmented to compute various volumetric measures, such as the size of different brain regions, cortical thickness, and surface area (Fischl & Dale, 2000; Ashburner & Friston, 2000). Specific regions of the brain, like the hippocampus, prefrontal cortex, and the gray matter volume, have shown correlations with age and are considered as biomarkers for aging (Raz et al., 2005).

### 4.3.1 RANK SELECTION

The scree plots for mode-2 and mode-3 unfoldings in Figure 4(b) and 4(c), respectively, show that after rank 7 the reduction in magnitude of the singular values is not significant. Similarly, the reconstruction error in Figure 4(d) also exhibits that the reconstruction error decreases slowly after rank 7. The scree plot for the mode-1 in Figure 4(b) diminishes quicker than the two other modes, however it does not reach another order of magnitude at rank 20. Considering the information provided by the scree plots and reconstruction error, we picked $R_1 = 7$ and $R_2 = 4$.

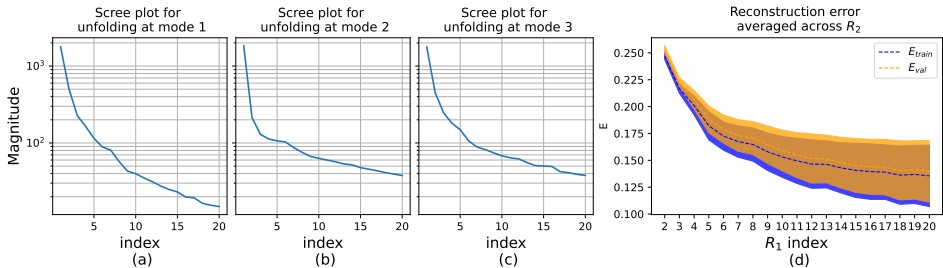

Figure 4: Scree plots and reconstruction error for MRI data. Scree plots for mode-1 $(a)$, mode-2 $(b)$ and mode-3 $(c)$ show a reduction in the magnitude of the singular values of almost one order of magnitude at rank 5. The reduction in singular values magnitude for mode-1 drops faster than magnitudes for mode-2 and mode-3. The reconstruction error plots $(d)$ for the validation and training sets show a considerable reduction at rank $R_1 = 7$.

### 4.3.2 SUBJECT-SPECIFIC LOADINGS $\mathbf{U}^{(1)}$, REGIONS OF INTEREST, AND AGING

Numerous studies have investigated the relationship between gray matter volume and aging, primarily utilizing MRI scans to quantify this relationship. Fjell et al. (2009) found that gray matter atrophy was correlated with age, especially in the frontal and parietal regions of the brain. Another study by Good et al. (2001) demonstrated a linear decline in gray matter volume with increasing age, particularly evident after the age of 50. Raz et al. (2005) used longitudinal MRI data to show that the decline in gray matter volume is a continuous process and that different brain regions may experience different rates of atrophy over time. These studies collectively underscore the significance of gray matter volume as a biomarker for both aging and various neurodevelopmental and neurodegenerative conditions.

The plots in Figure 5 explore the associations between age and loadings $\mathbf{U}^{(1)}$ related to brain volume. Interestingly, the trends in the plots exhibit a variety of patterns: Figure 5(b) and Figure 5(c), associated two RoIs 2 and 3, manifest a linear association between the loadings and age, Figure 5(f) and Figure 5(g), linked to RoIs 6 and 7, display a piecewise linear relationship, while Figure 5(a), Figure 5(d) and Figure 5(e), connected to RoIs 1,4 and 5, depict a non-linear pattern. A noteworthy observation is that the loadings corresponding to brain volume measurements exhibit heightened sensitivity in the younger population as compared to those obtained from stiffness measurements. This implies that brain volume may serve as a more robust indicator for characterizing neurodevelopmental changes during childhood and adolescence. Interestingly, all the relationships are linear for the population older than 50, which supports the results presented by Good et al. (2001).

As reported by Raz et al. (2005), various areas of the brain might undergo atrophy at distinct rates as time progresses. This can be seen in Figure 6, which shows a combined view of the regions of interest for brain volume. Since there is no predominant RoI in this view, we can infer that all of them play a role at different stages of development. Additionally, parietal and frontal lobes, known to be correlated with age(Fjell et al., 2009), are clearly identified by RoIs 4 and 5, respectively.

## 5 CONCLUSION

In summary, the paper has broken new ground by offering a distinct pathway for interpreting 3D convolutional neural networks applied in healthcare, specifically for brain maps. Unlike traditional image

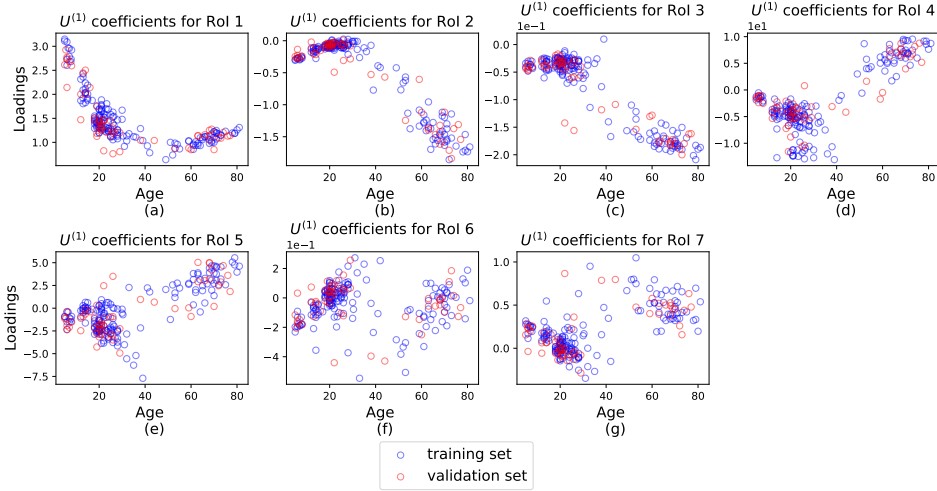

Figure 5: Loadings for RoIs. Relationships between age and loadings shows linear behavior for RoIs 2 and 3, and non-linear behavior for RoI 1, 4 and 6. Variations in volume are more sensitive for the younger population compared to stiffness.

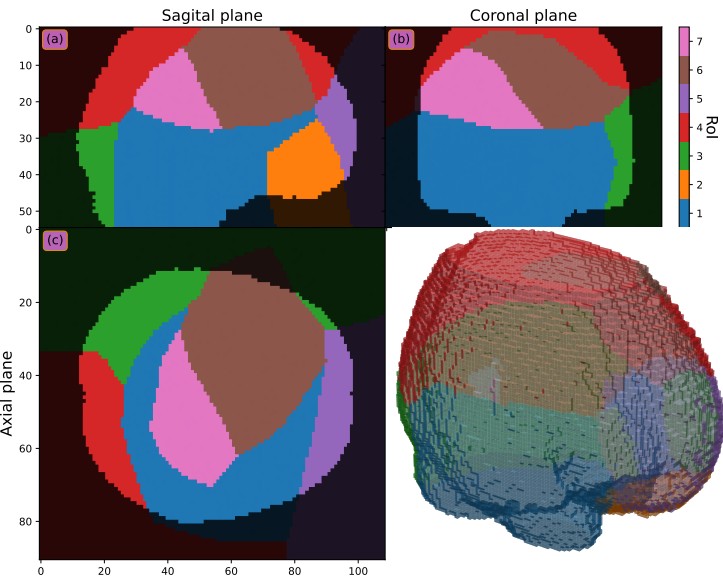

Figure 6: Combined Regions of Interest for Volume. Colored regions highlight which RoI has the strongest influence in a specific spatial location of the brain. Relationships are linear for the the population older than 50.

classification techniques like GradCAM, our approach leverages the common spatial registration of brain maps to deliver dataset-level explanations. By transforming the internal activations into a tensor format, we employ a constrained tensor decomposition to spotlight brain regions that the model considers critical during the prediction process. We applied this approach to models estimating an individual's age using MRI and MRE-derived brain maps. Notably, the areas of the brain highlighted by the tensor decomposition are consistent with those scientifically understood to be affected by aging. Moreover, a linear model fitted to the decomposed activations performs closely to the original model. This novel tensor decomposition method thus not only makes 3D convolutional neural networks more interpretable but also reveals valuable information about age-related changes in brain structure.

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

# A  APPENDIX

## A.1  ALGORITHMS

---

**Algorithm 1:** Rank evaluations for activations in training and validation set

---

**Input:** $\mathcal{A}_{\text{train}}, \mathcal{A}_{\text{val}}$
**Output:** $\mathbf{E}_{\text{train}}, \mathbf{E}_{\text{val}}$

1 **for** $(R_1, R_2) \in \{\{2, \ldots, 20\} \times \{2, \ldots, 20\}\}$ **do**

2 $\quad$ Solve:

$$\mathcal{G}_{\text{train}}, \mathbf{U}_{\text{train}}^{(1)}, \mathbf{U}_{\text{train}}^{(2)}, \mathbf{U}_{\text{train}}^{(3)} = \underset{\mathcal{G}, \mathbf{U}^{(1)}, \mathbf{U}^{(2)}, \mathbf{U}^{(3)}}{\text{argmin}} ||\mathcal{A}_{\text{train}} - \mathcal{G} \times_1 \mathbf{U}^{(1)} \times_2 \mathbf{U}^{(2)} \times_3 \mathbf{U}^{(3)}||,$$

3 $\qquad\qquad\qquad\qquad\qquad\qquad$ s.t. $\quad \mathbf{U}^{(3)} \geq 0, \ \mathcal{G}_{ijk} \geq 0$ for $i = k, \ \mathcal{G}_{ijk} = 0$ for $i \neq k$

4 $\quad$ Compute: $E_{\text{train}}^{(R_1, R_2)} = \frac{||\mathcal{A}_{\text{train}}^L - \mathcal{G}_{\text{train}} \times_1 \mathbf{U}_{\text{train}}^{(1)} \times_2 \mathbf{U}_{\text{train}}^{(2)} \times_3 \mathbf{U}_{\text{train}}^{(3)}||}{||\mathcal{A}_{\text{train}}^L||}$

5 $\quad$ Solve:

$$\mathbf{U}_{\text{val}}^{(1)} = \underset{\mathbf{U}^{(1)}}{\text{argmin}} ||\mathcal{A}_{\text{val}} - \mathcal{G}_{\text{train}} \times_1 \mathbf{U}^{(1)} \times_2 \mathbf{U}_{\text{train}}^{(2)} \times_3 \mathbf{U}_{\text{val}}^{(3)}||$$

6 $\quad$ Compute: $E_{\text{val}}^{(R_1, R_2)} = \frac{||\mathcal{A}_{\text{val}}^L - \mathcal{G}_{\text{train}} \times_1 \mathbf{U}_{\text{val}}^{(1)} \times_2 \mathbf{U}_{\text{train}}^{(2)} \times_3 \mathbf{U}_{\text{train}}^{(3)}||}{||\mathcal{A}_{\text{train}}^L||}$

7 **return** $\mathbf{E}_{train} = \left[ E_{train}^{(2,2)}, E_{train}^{(2,3)}, \ldots, E_{train}^{(20,20)} \right], \mathbf{E}_{val} = \left[ E_{val}^{(2,2)}, E_{val}^{(2,3)}, \ldots, E_{val}^{(20,20)} \right]$

---

---

**Algorithm 2:** Sparsity evaluations for RoIs

---

**Input:** $\mathcal{A}_{\text{train}}, \mathcal{A}_{\text{val}}, y_{\text{train}}, y_{\text{val}}, (R_1, R_2)$
**Output:** $\lambda_{\text{best}}$

1 **for** $\lambda \in \left\{ 1e^3, 1e^2, 1e^1, 1, 1e^{-1}, 1e^{-2}, 1e^{-3} \right\}$ **do**

2 $\quad$ Solve:

$$\mathcal{G}_{\text{train}}, \mathbf{U}_{\text{train}}^{(1)}, \mathbf{U}_{\text{train}}^{(2)}, \mathbf{U}_{\text{train}}^{(3)} = \underset{\mathcal{G}, \mathbf{U}^{(1)}, \mathbf{U}^{(2)}, \mathbf{U}^{(3)}}{\text{argmin}} ||\mathcal{A}_{\text{train}} - \mathcal{G} \times_1 \mathbf{U}^{(1)} \times_2 \mathbf{U}^{(2)} \times_3 \mathbf{U}^{(3)}|| + \lambda ||\mathbf{U}^{(3)}||_1,$$

3 $\qquad\qquad\qquad\qquad\qquad\qquad$ s.t. $\quad \mathbf{U}^{(3)} \geq 0, \ \mathcal{G}_{ijk} \geq 0$ for $i = k, \ \mathcal{G}_{ijk} = 0$ for $i \neq k$

4 $\quad$ Compute: $\hat{\mathcal{A}}_{\text{train}} = \mathcal{G}_{\text{train}} \times_1 \mathbf{U}_{\text{train}}^{(1)} \times_2 \mathbf{U}_{\text{train}}^{(2)} \times_3 \mathbf{U}_{\text{train}}^{(3)}$

5 $\quad$ Solve:

$$\mathbf{U}_{\text{val}}^{(1)} = \underset{\mathbf{U}^{(1)}}{\text{argmin}} ||\mathcal{A}_{\text{val}} - \mathcal{G}_{\text{train}} \times_1 \mathbf{U}^{(1)} \times_2 \mathbf{U}_{\text{train}}^{(2)} \times_3 \mathbf{U}_{\text{val}}^{(3)}||$$

6 $\quad$ Compute: $\hat{\mathcal{A}}_{\text{val}} = \mathcal{G}_{\text{train}} \times_1 \mathbf{U}_{\text{val}}^{(1)} \times_2 \mathbf{U}_{\text{train}}^{(2)} \times_3 \mathbf{U}_{\text{train}}^{(3)}$

7 $\quad$ Solve: $\mathbf{w}^\star = \text{argmin}_{\mathbf{w} \in \mathbb{R}^{c_L}} ||\hat{\mathbf{A}}_{\text{train}} \mathbf{w} - \mathbf{y}_{\text{train}}||_2^2$

8 $\quad$ Calculate: $MAE_\lambda = ||\hat{\mathbf{A}}_{\text{val}} \mathbf{w}^\star - \mathbf{y}_{\text{val}}||_1$

9 **return** $\lambda_{best} = \text{argmin}_\lambda MAE_\lambda$

---

