# OpenReview forum: "Interpreting Age Predictions from Brain Maps via Deep Neural Activations and Tensor Decomposition"
_ICLR.cc/2024/Conference — ICLR 2024 Conference Withdrawn Submission_

### Official Review · Reviewer_vyRz · 2023-11-05

**Soundness:** 3 good
**Presentation:** 3 good
**Contribution:** 2 fair
**Rating:** 3
**Confidence:** 4

**Summary:**

This paper introduces a new technique for interpreting activation maps from 3D convolutional neural networks (3D-CNNs) using tensor based factorization as an alternative to other methods used for image classification (eg Class Activation Maps- CAM, gradient based class activation maps Grad-CAM). This method is applied to a 3D CNN model trained to predict chronological age from brain maps of volume and stiffness computed from magnetic resonance imaging (MRI) and T1-weighted magnetic resonance elastography (MRE) scans, respectively.

**Strengths:**

Although not novel in itself, the idea of utilizing tensor decompositions to examine intermediate activation maps is interesting and principled. The paper is well written and presented, and the main idea is straightforward to implement and test for new use cases. The main idea could be a viable alternative to CNN interpretability for different applications

**Weaknesses:**

The evaluation aspect of the paper is rather weak for the following reasons:

(a) No comparison to baseline methods is presented
(b) The validity of interpretability maps is mainly based on consensus with existing literature and is more on the qualitative side
(c) No cross-comparisons are performed by verifying with experts in the field
(d) No bootstrapped comparisons are performed to evaluate the robustness and variability in the inferred maps
(e) It is unclear whether the learned maps are consistent across different age groups in the cohort

**Questions:**

1. Could the authors please elaborate on why they introduced the non-negativity constraint in the factorization? What is the biological significance of this assumption and why is it appropriate from the CNN architectural standpoint (does the assumption change when non-ReLU activation functions are used?)

2. Are the solutions obtained by the decomposition unique? If not- how does this affect the interpretability of the factors?

3. Can such methods be extended to non-convolutional architectures and non-spatially contiguous input data, for example graph neural networks

---

### Official Review · Reviewer_KNBM · 2023-11-09

**Soundness:** 1 poor
**Presentation:** 1 poor
**Contribution:** 1 poor
**Rating:** 1
**Confidence:** 4

**Summary:**

The article discusses the use of existing machine learning methods such as tensor decomposition, 3D ResNet, and more to analyze a downloaded dataset. However, the application of these methods is to estimate the so-called "brain age" from brain imaging, which has already been debunked. As a result, the article has little novelty or practical use and serves more as a machine learning toy example.

**Strengths:**

The manuscript lacks originality as the authors have only reused pre-existing concepts, and no significant strengths are identified. The authors should read a recent publication that debunks the "brain age" concept.

Vidal-Pineiro D, Wang Y, Krogsrud SK, Amlien IK, Baaré WF, Bartres-Faz D, Bertram L, Brandmaier AM, Drevon CA, Düzel S, Ebmeier K. Individual variations in ‘brain age’relate to early-life factors more than to longitudinal brain change. elife. 2021 Nov 10;10:e69995.

**Weaknesses:**

Lack of originality and toy "brain age" validation, which has been debunked already, as mentioned in the above comments.

**Questions:**

Why focus on "brain age" when more valid diagnostic labels exist?

---

### Official Review · Reviewer_Gsu3 · 2023-11-10

**Soundness:** 2 fair
**Presentation:** 1 poor
**Contribution:** 2 fair
**Rating:** 1
**Confidence:** 2

**Summary:**

The paper tries to find the latent embeddings for the last layer of a 3D CNN and relate the embeddings to specific features in brain ROIs.

**Strengths:**

I am not personally convinced by the idea.

**Weaknesses:**

1. The paper describes an algorithm without properly describing the intuitions behind each algorithmic step clearly. There is no theoretical analysis of the approach they propose. In the absence of proper theoretical guarantees, a thorough study on simulation data may benefit the paper.

2. The main contributions in the introduction section do not contain anything that makes the paper's contribution stand out from traditional approaches of finding the embeddings. I would carefully flesh out the novelties of the proposed approach.

3. Plots are not clear: why there is a spread of error in Figure 4 (d).

**Questions:**

1. The pseudocodes are not clear to me. How do you estimate $\mathbf{U}^{(3)}_{val}$ in Algorithm 1? The algorithm seems to switch a lot between estimates from training and validation sets for different embeddings? Why do you need that?

2. What is MAE in Algorithm 2?

3. The paper needs a complete rewriting to explain the intuition coherently and intuitively.

**Details Of Ethics Concerns:**

None.